# Participatory Landscape Conservation: A Case Study of a Seasonally Dry Tropical Forest in Michoacan, Mexico

**Neyra Sosa [1,2], Alejandro Torres [2], Valerio Castro-Lopez [2] and Alejandro Velazquez [1,3,*]**

1   Centro de Investigaciones en Geografía Ambiental, Universidad Nacional Autónoma de México, Antigua Carretera a Pátzcuaro 8701, Morelia 58190, Michoacan, Mexico; neyra.sosa@enesmorelia.unam.mx
2   Escuela Nacional de Estudios Superiores (ENES), UNAM Campus Morelia, Antigua Carretera a Pátzcuaro 8701, Morelia 56620, Michoacan, Mexico; alejandro_torres@enesmorelia.unam.mx (A.T.); clvalerio.11@gmail.com (V.C.-L.)
3   International Office, Freie Universitat Berlin, 14195 Berlin, Germany
*   Correspondence: alex@ciga.unam.mx; Tel.: +52-5521151977 or +49-17672607661

**Abstract:** Participatory landscape conservation is an innovative approach that weaves together theoretical models and practical applications. Intertropical regions, such as Mexico, face challenges to conciliate regional governability, social justice, and nature conservation. The State of Michoacan is one of these regions where such challenges are exacerbated, particularly nature conservation, due to ongoing territorial disputes. We implemented the participatory landscape conservation approach by creating a complementary form of protected area to deal with ongoing conflicts, drought conditions, and extreme poverty. We conducted participatory mapping and landcover/use analyses as the main methodological tools to reach consensus among stakeholders. We integrated, macro, micro, and social scales to provide sound arguments to integrate local, scholar, and policy makers' perceptions. The outcomes of the participatory mapping analyses were assessed. The present paper provides evidence of the positive outcome of using The Participatory Landscape Conservation Approach to establish a Biosphere Reserve, safeguarding one of the most biologically diverse and delicate ecosystems consisting of seasonally dry tropical forests within a rather disputed region. We discussed the relevance of our findings and compared them to ongoing regional and global trends in light of other forms of establishing long-term multistakeholder agreements, as is the case for protected areas.

**Keywords:** participatory science; biodiversity conservation; landscape science; Michoacan; Mexico

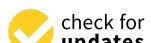



## 1. Introduction

### 1.1. Land-Based Conservation

An estimated one-third of the world's population relies on forests for subsistence, while more than two-thirds rely on resources and services derived from native vegetation areas [1]. Unfortunately, natural resources are dwindling rapidly, especially in tropical areas where community identity and culture are crucial to daily life. Such regions heavily depend on livelihoods derived from their ecosystems [2]. Protected areas (PAs) have long been considered a primary tool for preserving natural biodiversity. However, due to different cultures and contexts, the effectiveness of these areas has become contested in recent years. For example, some studies suggest that PAs may be instrumental in ensuring long-term conservation efforts [3]; nevertheless, other researchers argue that their failure to prevent deforestation in tropical regions is cause for alarm [4,5]. Reasons to explain these failures are place-based; yet, in most cases, engagement of local stakeholders has been neglected.

Additional research must be conducted to discover solutions that will safeguard the environment. Studies suggest that half of all PAs worldwide are inadequately managed, resulting in ecological upheaval, vegetation cover depletion, and plummeting endangered species populations [6]. Shockingly, in certain circumstances, ecological destruction increased after the protected area was created [7]. Therefore, several authors are requesting

new strategies to bolster PA performance, especially in tropical areas [4,8], as a means of assuring that socio-geo-ecological systems and livelihoods will endure within these territories [9]. In the 1970s, national parks developed a less restrictive biosphere reserves modality where local people were allowed to maintain their land tenure as long as their management actions were not jeopardizing conservation principles. Conservation should be done through interdisciplinary approaches [10–12] where scientific and local knowledge and political wills are evenly integrated [5,13–15]. In the face of our increasingly contested world, Bray and Velázquez [16] proposed that a vital landscape approach should be conducted to redirect public policy decisions and financing in line with sustainability principles. Landscape approach is an ever-evolving construct comprising interactions between natural and sociocultural components. It is regulated to meet human values, such as equity and development targets, with long-term environmental repercussions [17]. This approach aims at ensuring the sustainable utilization of existing resources while meeting societal objectives simultaneously.

### 1.2. Participatory Science and Landscape

As highlighted by Funtowicz and Ravetz [18], the outcomes of scientific studies must abide by governance principles, forming a bond between those involved in public/civil society/citizenship matters and their institutions with ruling bodies such as government entities, private sector organizations, and related establishments. Robust codes of conduct, accountability, and effectiveness should be established to ensure sound stewardship. Such management must also be participatory and comprehensive [19]. In resource management, at least two complementary conceptual frameworks have addressed these principles: namely, the polycentric governance [20] and the collective impact [21]. The first states its fundaments in constructing bottom-up common interinstitutional actions, whereas the latter focuses on identifying multistakeholder initiatives. The approaches of Ostrom [20] and Kania and Kramer [21] formulate a route and principles to eventually find a way out to conciliate long-term commitments, and to engage stakeholders into developing practical solutions that simultaneously address the territory's biophysical constraints and fulfill its socio-cultural expectations. Furthermore, this negotiation process is essential to effectively mediate conflicting interests on the landscape. Therefore, emphasis is placed on "pluralism" in negotiated landscapes where the spatial context has been largely ignored as the common ground [22,23]. Participatory science is best illustrated in the "strict national park" concept where local stakeholders have been around since the beginning of these areas' conservation efforts and may still reside within them, so that asserting their right to participate actively in their management becomes crucial [16].

### 1.3. Geopolitical Context

Despite representing a vital global biodiversity reservoir [24,25], tropical and intertropical countries, such as Mexico, experience rapid deforestation [26,27]. Mexico, as most countries worldwide, rely on protected areas as a means to conserve their native genetic asset. In Mexico, 185 protected areas have been established to protect biodiversity. These PAs cover 90,958,374 hectares (46.5% of the national continental and marine territory), only 11% of which is continental [28]. Many PAs have been evaluated as nonfunctional in their decree objectives [29]. In Mexico, land ownership consists of public properties that belong to the nation, individual private possessions termed small property, and ejidos and indigenous lands. These last two are classified collectively as social property or agrarian communities. Unique to Mexico, agrarian communities result from historic agricultural reforms in 1934 and 1992 that created separate forms of land ownership. As a result, a massive 102 million hectares of Mexican land are dedicated to two distinct types of property—ejidos, comprising 84.5 million ha, and indigenous communities, with 17.4 million ha. This accounts for 53.4% of Mexico's total land surface [30]. Mexico is the global leader in communal forest enterprises, with more than 80% of its forests managed by local stakeholders [31]. The highest governing body of ejidos and rural communities in Mexico is the general assembly,

comprised of a commissioner, secretary, and treasurer, who ensure effective management. More than 5.6 million commoners and owners raise numerous products for family use and to meet national demand—crops, livestock goods, and fodder—in more than 34,000 ejidos and communities in Mexico. They also manufacture construction materials, handicrafts, tourist services, and other items suitable for international purchase [32]. This natural asset is an integral part of the nation's capital. It provides invaluable services and resources, including its unparalleled biodiversity, carbon sequestration capacity, groundwater replenishment capability, supportive ecosystem functions, regulations, and cultural heritage [33]. To our knowledge, there is scanty research that integrates political and social stakeholders to accomplish valuable long-term allies in biodiversity conservation on regions with ongoing territorial disputes [34].

### 1.4. Objectives

The aim of the present paper is threefold. Our primary goal was to develop an active implementation of participatory landscape conservation and use it to create a system of conservation areas in the State of Michoacan. Our second goal was to apply our initial achievement by creating a complementary form of protected area to ensure maximum protection while improving marginalized communities' lives. The third objective was to evaluate the success of the complementary form of protected area fifteen years after its establishment.

## 2. Methods

### 2.1. Study Area

The research took place in the State of Michoacan, which covers 58,599 km$^2$ of mostly mountain-dominated landscape (see Figures 1 and 2 for a map of the study area, highlighted in black). It contains portions of the Trans-Mexican Volcanic Belt (formed in the Quaternary period) and the Sierra Madre del Sur (formed in the Paleogene period). Climate varies depending on the elevation and geographic emplacement (coastal-to-inland). In this State, Nearctic and Neotropical biogeographic realms converge. As a result, Michoacan harbors outstanding biodiversity (e.g., 845 tree species), with about 40% of the species listed as endemic and/or threatened. It is comprised by 113 municipalities and about half of its present area is governed by agrarian communities. Gopar-Merino et al. [35] have provided a critical review of the biophysical complexity of Michoacan, which was referred to as an outstanding ecogeographical complex macroregional state. A social profile is given in the next sections.

### 2.2. Macroregional State Level

In consensus with the Michoacan State authorities, in between 2005–2007, we conducted a state-level consultation by active participatory workshops aimed at twofold goals: (1) identification of priority areas of environmental, social, and economic importance; (2) delineation of a consensual conservation strategy. The primary sources for the active participatory workshops were maps depicting abiotic (geology, landform, and soils), biotic (biodiversity), and land tenure. The main source was the Mexican Mapping Agency (INEGI is its Spanish acronym). Furthermore, remote sensing tools such as satellite images and aerial photographs were used in conjunction with relational databases to produce maps showcasing population size and marginalization across the state territory, as well as vegetation and land use, deforestation processes, human settlements, industrial corridors, and environmental management policies. Six workshops were conducted with three types of stakeholders: namely, five with agrarian communities (most importantly, with their authorities in turn), and one with scholars and representatives from the federal, state, and municipal governments.

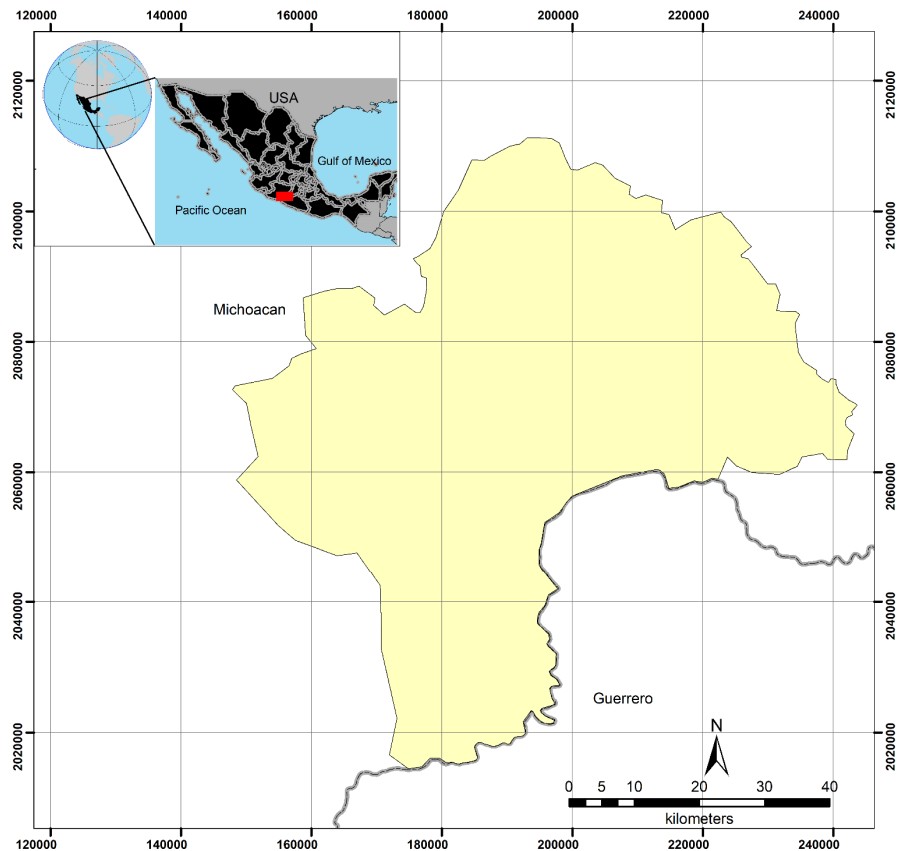

**Figure 1.** Location of the study area. On the tope left side of the of the figure Mexico is emplaced, and in red the State of Michoacan.

Due to the extent of the macroregion and the complexity for logistics, the State of Michoacan was split into five regions on the basis of accessibility and positive neighbor relationships for workshops with agrarian communities. Each of these followed three stages: first, the state governor of Michoacan issued a call-to-action; second, the Ministry of Urbanism and Environment (SUMA is its Spanish acronym) handled logistical matters; third, authors and local authorities worked together to implement the consultation process. Participants were organized into tables (of about ten to fifteen people) where maps were overlaid, covered with acetates. On their maps, participants delineated areas of socio-environmental value. After the full-day workshop, partial results of each table were presented in a collective forum. During this presentation, agreements were made on proposing protecting certain areas for conservation without jeopardizing ongoing future development projects.

The sixth workshop was attended by scholars from various backgrounds, including the natural, social, and humanities sciences, in one room split into interdisciplinary tables. Simultaneously, in another room, representatives from municipal, state, and federal government entities also conducted the same exercise. This workshop featured the same components as its regional counterparts, although with a heightened focus on delineating agreement among areas of immense socio-environmental merit. To maximize the effectiveness of this sixth workshop, a minimum mappable area was determined (100 hectares for maps at 1:250,000 scale). Additionally, preliminary data on climatic variability e.g., [35], biological richness e.g., [36], and vegetation diversity e.g., [37] were provided in combination with geographical proximity to production systems (e.g., avocado plantations) and human settlements. At the end, both groups from the two rooms were gathered together to review their outcomes collectively.

The outcomes of the six workshops were integrated using a Geographic Information System by overlapping all delineated areas on a raster map of cells of one squared kilometer each. Each cell (pixel) was given a weight according to the number of times it was selected by one of the stakeholders. Cells with less than three nominations out of the six workshops were not included in the second phase of the integrated analyses. In the second phase, assessment of contiguity, connectivity, and fragmentation was computed so that the cells that were most isolated (total distance to the next group of cells) and small (number of cells clustered together) were also pondered as a second priority. This preliminary second phase's weighted outcome was presented to the municipal, state (Governor and Minister of Environment of the State), and federal authorities (National Commissioner of Protected Areas of Mexico), so that a final decision was made to define a so-called State System of Conservation (SSC). Policy makers pointed out that one of the areas of the SSC located in the tropical dry ecosystem was to be further evaluated for its social, cultural, environmental, and political relevance.

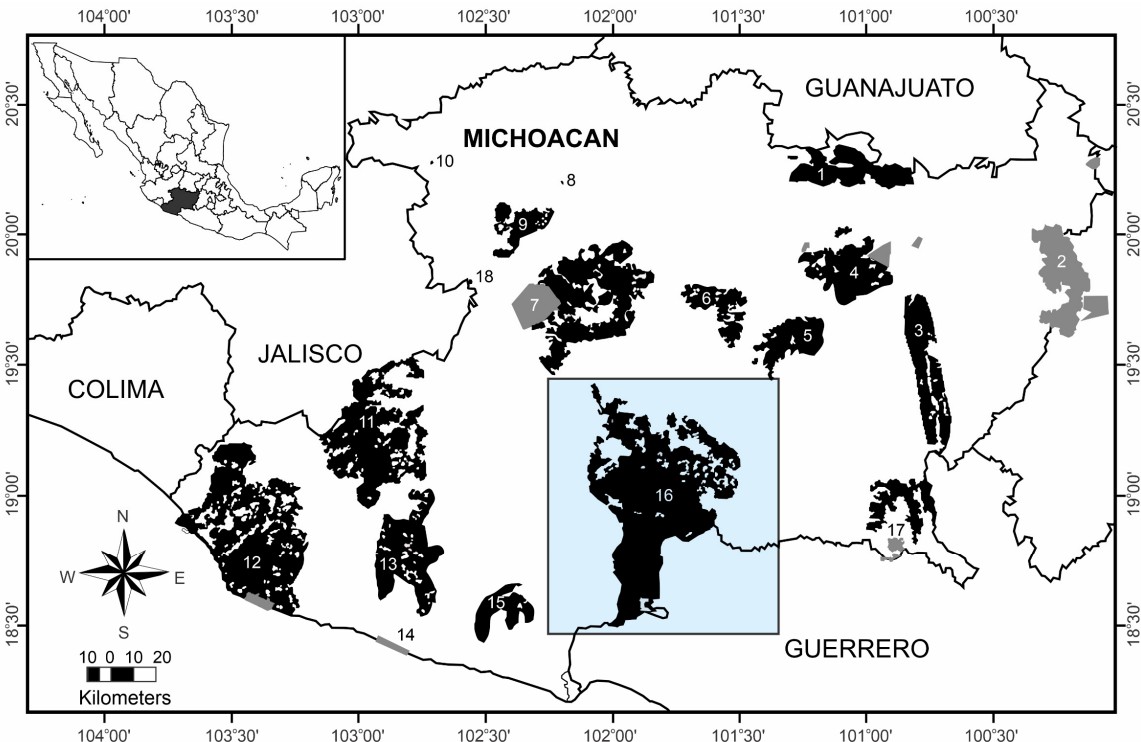

**Figure 2.** The top-left corner contains the map of Mexico where the State of Michoacan is highlighted in black. The overall map contains the reconciled areas from consultation among civil society and rural communities, academic circles, and government institutions. In grey color, the polygons of the current protected areas are established. Area number 16 became a priority because of its biocultural nature, and it was chosen as the target area to explore further participatory conservation. The frame around area number 16 corresponds to the microregional level. Polygons shapes between the current area number 16 and the Zicuirán-Infiernillo Biosphere Reserve differed as a result of negotiations at the local level.

### 2.3. Microregional Level

The zone numbered 16 (Figure 2) on the SSC map was pinpointed by the state and federal authorities as the region to further explore in terms of its suitability for establishing a protected area. This region, Zicuirán-Infiernillo, is one the most diverse and extended tropical dry forests; it faces high social complexity and governability and is regarded as vulnerable to climate change. The Zicuirán-Infiernillo region comprises parts of the Huacana, Arteaga, and Churumuco municipalities, and most of the Infiernillo Dam, which produces about 25% of Mexico's electricity out of all hydroelectrical dams [38].

To organize the public consultation in the assemblies of the agrarian communities, an intergovernmental group was formed by Arteaga, Churumucao, and La Huacana City Council members, five state government entities led by the Ministry of Environment of Michoacan, the National Commission of Protected Areas, and the authors of the present paper. The group held seven meetings to discuss how to present, disseminate, and eventually engage civil society, agrarian communities, and non-government organizations (NGOs) Three steps were considered prior to the consultation:

(1) Enrollment of active NGOs that have played an important role in making local inhabitants aware of their land's natural values (e.g., The Community Biodiversity Conservation Program, the Project for the Conservation and Sustainable Management of Forest Resources, Bajo Balsas Non-Governmental Organization).

(2) Preparation of detailed cartography at a medium scale (1:100,000 and 1:50,000) to illustrate the interconnectedness of the agrarian community's lands with various basins and sub-basins (water is a critical resource in the region), landcover, land use, human settlements, primary and secondary roads, and boundaries of agrarian communities.

(3) Planning open public consultations to include small landowners, experienced technical service providers, ejido counselors, and livestock associations.

The consultation process took place from February to July 2007, and it was conducted in presentations in general assemblies of the 64 agrarian communities identified with legal jurisdiction within Zone 16 of the SSC. Due to boundary disputes, the National Agrarian Registry's boundaries were not displayed on maps during presentations in assemblies when two or more agrarian communities were participating. The goals of each assembly focused on approval for adding an agrarian community as part of the ongoing construction of the Zicuirán-Infiernillo Biosphere Reserve (hereafter, ZIBR), as well as exploring their willingness to become part of the core zone (area uniquely used for biodiversity conservation purposes). Agreements of the assemblies were stated in minutes (official debriefings) so that collective decisions were backed up legally.

*2.4. Efficiency Assessment of the Zicuirán-Infiernillo Biosphere Reserve*

To assess the efficiency of the ZIBR, we conducted landcover/use change analyses by crossing two databases of different years (2005 and 2021). The established polygon of the ZIBR and its peripheral (buffer) zone (an adjacent area delimited by the National Commission of Protected Areas) were combined to assess the regional landcover/use trends.

We used, as baseline, (database T1) the National Institute of Statistics and Geography (INEGI) series III of 2005 (scale 1:250,000) as the year just previous to the establishment of the ZIBR. T1 database was constructed by the visual analysis of Landsat 7 images and comprised land use and vegetation formation classes. The labels used for these classes and their distribution patterns were confirmed during on-site inspections in 2007 and supplementary aerial images. A thorough description of the integration, correction, and compilation of the T1 database was given by Cuevas and Mas [39].

The T2 database featured vegetation formations (scale 1:100,000), obtained from the automated classification of SPOT images from 2018 and further verified through field research during 2020 and 2021, which included sampling tree species, according to Velazquez et al. [37] and Rangel-Landa et al. [40]. A scale of 1:250,000 was used to ensure that the two databases (T1 and T2) were compatible. Additionally, the minimum mapping area was set to be at least one km$^2$; thus, all polygons smaller than one km$^2$ had to be merged with the largest adjacent polygon for compatibility.

We reclassified T1 and T2 databases into three distinct cartographic classes: temperate dry forests, tropical dry forests, and cultural land use types. This latter class included crops, settlements, and livestock grazing areas where native vegetation was not predominant. Water bodies were kept as one stable landcover. We overlapped T1 and T2 databases by layering them onto a geographic information system and analyzing shifts and patterns across different periods following the procedure described by Velázquez et al. [41]. We

then computed the yearly rate of changes among classes by using the method described by Velázquez et al. [26].

## 3. Results

### 3.1. The State System of Conservation (SSC)

At the macroregional level, two hundred and ninety-eight people attended the six workshops and 2659 surveys were collected from those who could not participate. After executing surveys and workshops, we mapped out 18 initial areas, covering 10,399 km², or about 18%, of the landmass of Michoacan (Table 1). The SSC surpasses the combined federal and state protection efforts by ten times (Figure 1). This result combines bottom-up and top-down participatory processes, where social actors are the catalysts for defining, limiting, and managing potential regions to become protected areas.

**Table 1.** Eighteen areas were determined through a consensus of 95% agreement between the three social sectors participating in consultations and workshops. The Protected Areas column denotes those that have been legally set aside and encompass, to a full or partial extent, the objectives of this academic exercise.

| Number on Map | Areas | The State System of Conservation | | Established Protected Areas by 2014 | |
|---|---|---|---|---|---|
| | | Surface (Km²) | % | Surface (Km²) | % |
| 1 | Cuitzeo-Copandaro | 421.52 | 0.71 | 2.54 | 0.02 |
| 2 | Monarch Butterfly Biosphere Reserve | 562.79 | 0.95 | 562.79 | 5.37 |
| 3 | Tiquicheo-Tzitzio-Madero | 546.14 | 0.93 | 0.00 | 0.00 |
| 4 | Morelia-Tzitzio | 540.64 | 0.92 | 66.59 | 0.64 |
| 5 | Madero-Tacambaro | 317.19 | 0.54 | 0.77 | 0.01 |
| 6 | Opopeo | 244.18 | 0.41 | 0.00 | 0.00 |
| 7 | Pico de Tancítaro | 1193.98 | 2.02 | 222.22 | 2.12 |
| 8 | Parque Nacional Lago de Camecuaro | 0.11 | 0.00 | 0.11 | 0.00 |
| 9 | Los Reyes | 206.49 | 0.35 | 0.00 | 0.00 |
| 10 | Parque Juárez de Jiquilpa | 0.08 | 0.00 | 0.08 | 0.00 |
| 11 | Coalcoman | 1110.34 | 1.88 | 0.00 | 0.00 |
| 12 | Chinicuila-Coahuayana | 1615.54 | 2.74 | 33.94 | 0.32 |
| 13 | Aguililla-Coalcoman-Tumbiscatio | 649.26 | 1.10 | 0.00 | 0.00 |
| 14 | Playa Mexiquillo | 31.35 | 0.05 | 31.35 | 0.00 |
| 15 | Arteaga | 241.73 | 0.41 | 0.00 | 0.00 |
| 16 | La Huacana-Churumuco-Artega | 2418.77 | 4.10 | 0.00 | 0.00 |
| 17 | Huetamo-Turitzio | 298.40 | 0.51 | 0.00 | 0.00 |
| 18 | Chorros del Varal (Los Reyes) | 0.73 | 0.00 | 0.73 | 0.01 |
| Total | | 10,399.24 | 17.63 | 921.13 | 8.79 |

Our research into participatory landscape conservation unveiled the fact that eight of the eighteen designated territories (illustrated in Table 1: 3, 6, 9, 11, 13, 15, 16, and 17) had never been taken into consideration for conservation. The Monarch Butterfly Biosphere Reserve (No. 2) and the Pico de Tancítaro Flora and Fauna Protection Area (No. 7), both temperate ecosystems, are currently at the heart of highly contested social disputes. Despite their ecological relevance, numbers 8, 10, 14, and 18 were relatively small areas to be considered as priorities at the state level. Numbers 11, 12, and 13

comprised outstanding biodiversity, yet these are currently ongoing social disputes, so environmental considerations are not at the top of the agenda for municipal, state, and federal governments.

### 3.2. Zicuirán-Infiernillo Biosphere Reserve Consultation

A total of 115 assemblies were conducted in six municipalities and 64 agrarian communities with the participation of 1999 ejidatarios (members of the agrarian communities with legal rights for land tenure). Sixty out of the 64 outvoted to support the creation of a new biosphere reserve with signed assembly minutes. Out of the 60 agrarian communities, only 26 have agreed on establishing a portion of their land as a core zone, which implies no human action other than biodiversity conservation. For a comprehensive overview of the rural communities' name, municipality, proposed and agreed-on core zones, and mean of agreement, please refer to Table S1.

After a thorough assessment, it was decided that 265 thousand hectares of land should be allocated in the Arteaga, Churumuco, Huacana, and Tumbiscatío municipalities. This area would encompass four core zones, spanning 22 thousand hectares and an additional 189 thousand hectares as a buffer zone. Sixty agrarian communities and 134 small owners joined this conservation proposal. On 30 November 2007, the Zicuirán-Infiernillo region was officially established as a Biosphere Reserve [42].

### 3.3. Biosphere Reserve Model Efficiency

In 2005 (T1), most of the region was covered by tropical dry forest (71.56% or 317,888 hectares). Cultural land use types accounted for 19.77%, while temperate dry forest comprised 4.79%. By 2021 (T2), the tropical dry forest had significantly increased its surface by 10%, expanding to 360,781 hectares (81.22%). On the other hand, cultural land use declined to 48,202 ha, accounting for 10.85%, whereas temperate dry forests almost remained the same, with changes accounting for less than one percent (Figure 3).

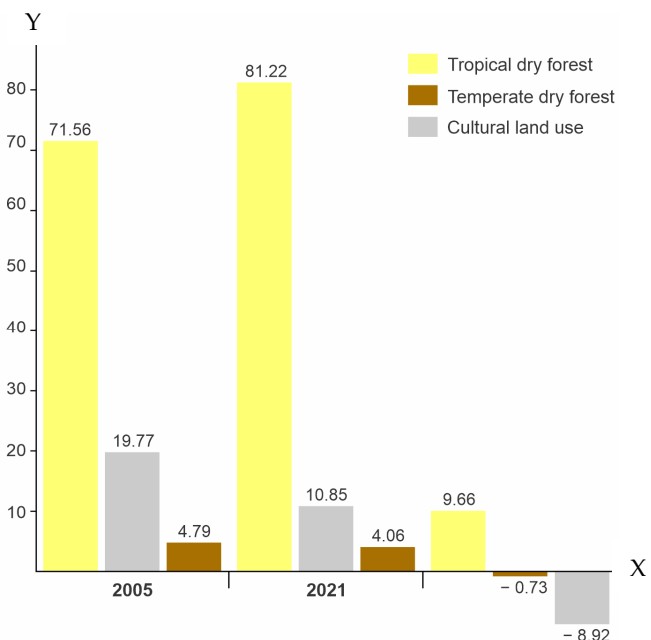

**Figure 3.** Conversion data among landcover classes. The "Y" axis states for percentage of landcover class in relation to the whole region; whereas the "X" axis accounts for the years of assessment. The right part of the figure reports the percentage of the landcover class that either gains or loses surface when comparing a landcover class between 2021–2005. Tropical dry forests have increased by about 10% on their surface over 15 years within the Zicuirán-Infiernillo Biosphere Reserve and its buffer zone. Most of the increase occurred due to cultural land use, whereas changes in temperate dry forests have been negligible.

The participatory landscape conservation approach allowed us to reveal spatially explicit conversion processes (Figure 4) expressed in annual rates of change (Figure 5). Protected area establishment, however, may not be held accountable for these results alone. Factors such as territorial disputes, outmigration, and extreme drought effects have all contributed, although these have not been thoroughly studied yet.

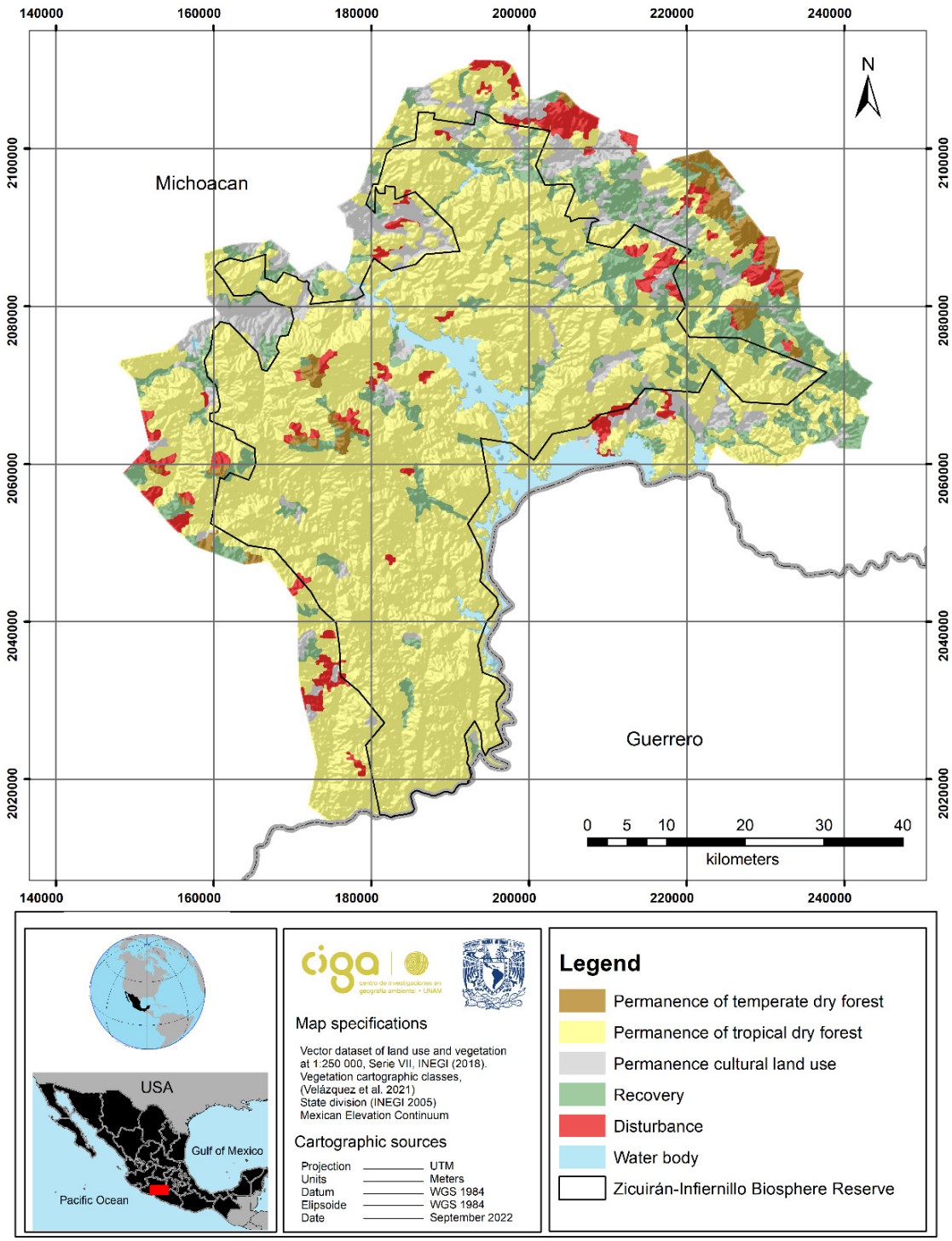

**Figure 4.** Spatially explicit conversion processes occurred in the entire region and within the Zicuirán-Infiernillo Biosphere Reserve. The green areas depict polygons where recovery from cultural land uses turned into tropical dry forests, in contrast to red polygons labeled as Disturbance, where the opposite landcover change occurred [41].

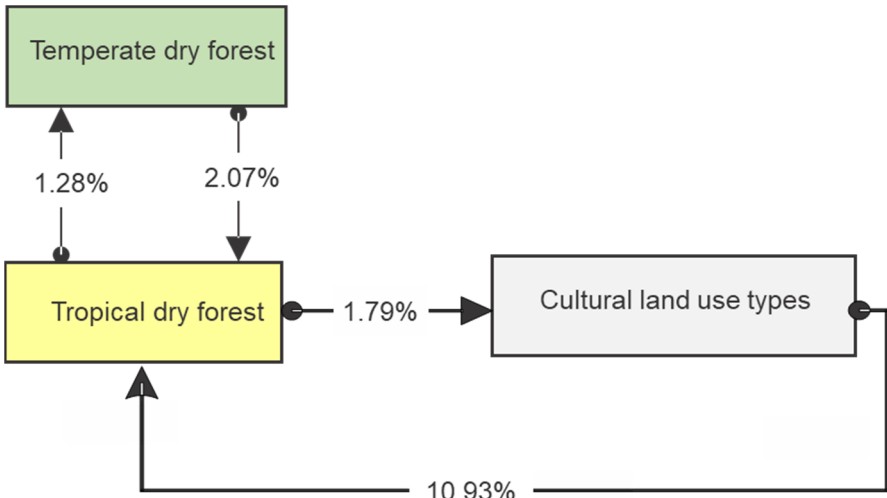

**Figure 5.** Land use transition matrix (T1 = 2005, T2 = 2021). Annual rates of change depict yearly transformation trends from one class to another. Conversion between forest types is relatively stable compared to the recovery speed observed in converting cultural land use types into tropical dry forests. Values below one percent were regarded as negligible.

The changes depicted in the conversion processes map (Figure 4) were field-cross-checked with the aid of the director of the protected area (Hugo Zepeda). The current maps helped him to share with the rural communities to find triggers of positive or negative trends. Transition trends were also calculated, as shown in Figure 5, where the annual rate of changes is indicated. This information was crucial for managing the protected area because transition matrices were requested per municipality to design sound land-based oriented public policies. These include different incentives for those rural communities that have promoted the recovery of the native tropical dry forests in contrast to the ones that have not.

## 4. Discussion

### 4.1. Multi(scale)stakeholder Integration

In the present paper, we developed the Participatory Landscape Conservation Approach (PLCA) as a complementary framework for the ongoing proposal for participatory science, governance, and collective impact [18,20,21]. The PLCA focuses on using the "Land" as the negation core for non-accountable ecosystem services such as carbon sequestration, water, biodiversity, and sociocultural attachment, here referred to as "The Bounded Heritage". The PLCA's conceptualization was developed simultaneously as we were learning how to implement it. The principles for sound implementation comprise six levels of imbrication, multistakeholder participation, scale perceptions, governance bodies, and, most importantly, decision drivers (Table 2). Levels I to III are place-based and socio-cultural dependent and are the most important for negotiations actions, whereas levels IV to VI are politically and economically driven and concern institutional -administrative forces.

Cross-cutting environmental problems would profit from a holistic framework when dealing with common resources such as Sustainable Development Goals [43]. In this line of thought, the PLCA may be regarded as a rural innovation development action where the land is placed at the center (Figure 6). Our conceptual PLCA nested model departs from a holistic perspective; it assumes complementary, sometimes contested, perceptions, recognizes the contrasting driving forces of each layer of stakeholders, and implies a larger degree of management uncertainty as we move from levels I to VI. Extensive research has called for a joint conceptual, albeit integrated, approach to enhancing socioecological system resilience, which includes recognizing problems, brainstorming solutions with stakeholders, assessing responses, and making modifications as needed [44]. Yet, implementation of integrated conceptual, albeit practical, approaches is scarce in scientific literature.

**Table 2.** The Participatory Landscape Conservation Approach (PLCA) nested model comprise six levels where stakeholders, scale perceptions, governance bodies, and decision drivers differed significantly among them. Scale is geographically, as well as socioculturally, relevant. States, nations (administrative countries or indigenous nations), and macroregions (e.g., the European Union, Mercosur, and others) keep changing, since land, on the whole, has always been contested.

| Level | Stakeholder | Scale | Governance Body | Decision Drivers |
|-------|-------------|-------|-----------------|------------------|
| I | Peasant | Meters | Family hold | Livelihood productivity |
| II | Agrarian community | Hectares | General Assembly | Territorial services |
| III | Municipality | $km^2$ | Mayor | Territorial control |
| IV | State | Hundreds of $km^2$ | Governor/State Minister | Socioeconomic policies |
| V | Nation | Thousands of $km^2$ | President/Minister | Political interests |
| VI | Macroregion | Millions of $km^2$ | Commissioner | Macroeconomic interests |

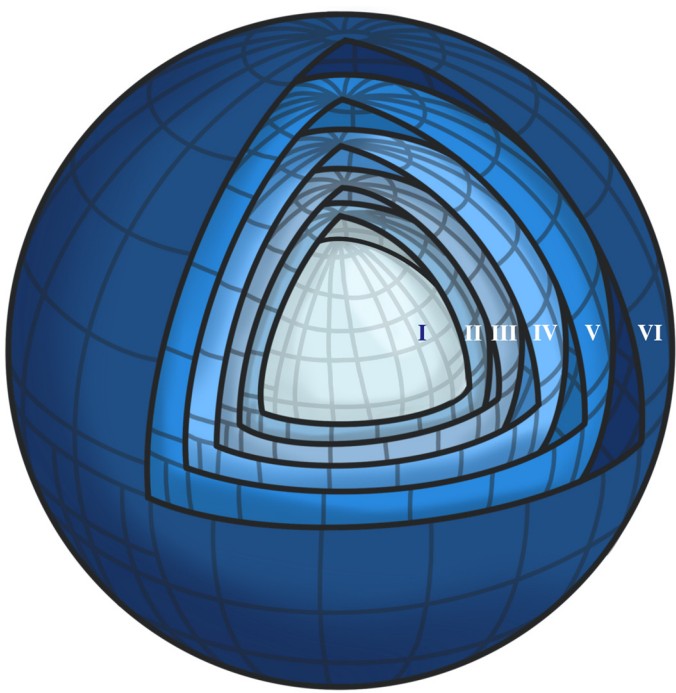

**Figure 6.** Revealing the comprehensive and holistic concept of the participatory landscape conservation approach. Land (Level I) is a core element to drive negotiations across multi-stakeholders. Each level´s attributes (I to VI) are thoroughly described in Table 2.

The State System of Conservation for Michoacan is derived from a participatory exercise that brought environmental perceptions to levels I to V of the PLCA nested model. The Michoacan governor at the time (Lázaro Cárdenas Batel) and his team understood the need to develop an extensive consultation. The leading participation of public universities (in this case, Universidad Nacional Autónoma de México and Universidad Michoacana de San Nicolás de Hidalgo) provided trustable grounds to have everyone on board during workshops. One research reveals the relevance of the neutral ground of the call made by universities, as happened in Aguililla Municipality, where even violent organized groups could express their views since they had interest in traffic control in specific areas, so small polygons were consensually appointed as relevant for conservation without jeopardizing local interests. Agrarian communities delineated small, specific, well-located areas. In contrast, scholars, knowledgeable about the natural richness of Michoacan, insisted on selecting large conservation areas so that integral biocultural attributes may be protected. As a result, scholars delineated about 70% of the whole surface of the State of Michoacan.

Overall, overlapping common interests on maps became a powerful negotiation tool so that all "holders" became aware of the 18 areas depicted as potential for biocultural conservation policies.

Regional participatory experience in Huacana, Churumuco, and Arteaga municipalities was initially considered a burden. The first assemblies resulted in disputes among participants (peasants), sometimes claiming rights over their neighbors. At the local scale, people believe their area is more significant and affluent than their neighbors'. To avoid that, local maps at the agrarian community level were prepared so that no comparisons could happen during assemblies and workshops. Nonetheless, 115 assemblies to engage 60 rural communities were needed to establish the Zicuirán-Infiernillo Biosphere Reserve (ZIBR) in Michoacan. This became a powerful platform to protect, conserve, and manage its natural resources. The abundant tropical dry forests in the ZIBR are a richly diverse ecosystem of many endemic species at risk due to human interventions. Hugo Zepeda, Director of the Zicuirán-Infiernillo Biosphere Reserve, commented recently, "the outcomes of the participatory approach have been remarkably positive and striking, and this area has proven resilient in the face of significant disruptions". Utilizing the participatory landscape conservation approach, peasants, local governments, producer organizations, and land management groups could join forces to achieve a unified regional goal. The last yearly assessment (2022), conducted by Hugo Zepeda Castro, concluded that a synergistic effect exists between encouraging people to abandon agricultural lands and subsequent recovery of dry tropical forests. "All agrarian communities are different, yet these share two situations: government disruptions due to organized crime taking over critical spaces, combined with a lack of support when faced with extreme weather conditions that adversely affect the productivity of their operations. Land fallows are then not always a result of pure environmental concern, nor the participatory scheme followed. Main proximal driven forces to explain the sound conservation and increase of the native tropical dry forests in the ZIBR included government subsidies for conservation actions, proactive participation of Grupo Balsas NGO, and engagement of universities for monitoring biodiversity. In contrast, underlying forces reveal that criminal organizations have established territorial and administrative control up to level III. Control by criminal organizations comprises two actions: peasants must be granted permission for agricultural and extensive livestock expansion and access to resources; and poachers are no longer allowed. Nowadays, the "Sembrando Vida" (https://programasparaelbienestar.gob.mx/sembrando-vida/ (accessed on 31 May 2023)), the new national policy targeted at supporting peasants to engage them in productive rural landscapes, seems to be offering positive results; however, the extent of the impact of the criminal organizations controls and the "Sembrando Vida" program are yet to be ascertained.

### 4.2. From State to National Scale

In Mexico, as in most hot-spot countries, the PLCA seems promising for melding together ideas and perspectives by stakeholders to formulate and execute environmental public policies. This strategy aims at engaging local players as allies in protecting their bounded heritage; thus, their land holds more cultural and environmental values. This strategy was crafted to prevent social problems from being implemented and managed without prior discussion [45]. A legitimate validation process needed to occur due to the constant territorial disputes in Michoacan. We can illustrate this with the Mexican Monarch Butterfly Biosphere Reserve, where academics and conservationists were the ones behind its establishment (Levels III to VI of Figure 6). However, local actors and agrarian communities (Levels I and II) were not on board with the original initiative, and current disputes persist despite the biological importance and outstanding budget allocated, even from international sources, as is the case of WWF-Mexico. At "El Vizcaíno" Biosphere Reserve in Baja California Sur, researchers concluded that its destiny relies upon a consensual governance regime. This reserve comprises the unique natural capital of the Baja California desert ecosystems, but level I to III stakeholders were ignored or partially

included throughout website opinions. According to Brenner and De la Vega [46] and Rosete et al. [47], the concept of a Biosphere Reserve can be relatively inclusive with significant potential for success. However, the redefinition of participation must be reviewed e.g., [48]. Mexican authorities launched an internet consultation before establishing a new protected area. Government consultation disregards that most local rural communities are not connected to the Internet, so regional agrarian conflicts are recurrent. The Monarch Butterfly and El Vizcaino Biosphere Reserves, as is the case of many others in Mexico, ought to be redesigned by following the PLCA so that all stakeholders reach commitment and engagement in the long-term.

According to Kolb et al. [49], solutions for sound environmental policies must be intricated due to the multi-leveled scope of institutional and geographical elements when approaching issues holistically. Thus, forming alliances and agreements is essential to establish collaborations and interventions [33,50]. According to López-Martínez and Cuanalo de la Cerda [51], a professional extensionist can be instrumental in strengthening the agrarian community's ability to come together and successfully handle any identified disputes. Salas et al. [52] analyzed the participation in conservation activities between two neighboring communities in Baja California over ten years. Surprisingly, they found that prior experience with travelers and tourist-related development agents and temporary migration to vacation spots fostered engagement in sustainability practices and the launch of community initiatives to safeguard marine areas essential for fish reproduction. A digital atlas was created to evaluate resilience and formulate plans by actively engaging the community in research. In addition, local leaders were trained on how to use this resource effectively.

*4.3. (Inter)tropical Outreach*

(Inter)tropical nations shared three outstanding features: they are largely inhabited by indigenous and mestizo ethnical groups [53], are rich in biodiversity and endemicity [54], and are contested territories [55]. Within these cultural, social, and political contexts, the framework of participatory landscape conservation approach adds a pragmatic path for the previous proposals of Funtowicz and Ravetz [18], Ostrom [20], and Kania and Kramer [21], and seems suitable for constructing bottom-up top-down interinstitutional and multistakeholder territorial initiatives. By relying on the PLCA, meaningful progress can be made toward boosting landscape resiliency. With imminent risks to food security and sovereignty, human health, biodiversity conservation, and ecosystem services in mind, the six levels of stakeholders ought to be regarded as allies, yet special attention must be placed in agrarian indigenous and mestizo communities since long-term management relies on weekly or monthly decisions. We must bear in mind that environmental public policies detached from engaging social stakeholders are meaningless. Climate change, One Health, social security, education, cultural identity, and territorial governance are closely connected to the environment [1,2,9]. Therefore, constructing effective environmental solutions requires a holistic place-based perspective that considers these aspects for eventual conservation success on The Bounded Heritage. Overlooking this complexity implies a misjudgment of human understanding, yet articulation remains challenging in the face of new geopolitical realities.

**Supplementary Materials:** The following supporting information can be downloaded at: https://www.mdpi.com/article/10.3390/land12112016/s1, Table S1: The table includes each agrarian nucleus's name, municipality, assemblies' dates, and meetings held to discuss core zones and agreements. After reviewing the 64 nuclei, four ultimately chose not to join the Biosphere Reserve by collective decision.

**Author Contributions:** N.S. conceived the research framework, collected data, conducted social consultation, wrote a preliminary manuscript, and led the field research. A.T. contributed to the research framework, collected data, and conducted social consultation. V.C.-L. performed statistical and geographical analyses and reviewed the last versions of the present paper. A.V. conceived the research framework, collected data, performed statistical analyses, wrote the paper, and led the contribution of all authors. All authors have read and agreed to the published version of the manuscript.

**Funding:** Financial support came from Universidad Nacional Autónoma de México (Project: DGAPA-PAPIIT IN105721).

**Data Availability Statement:** The data presented in this study are available on request from the corresponding author.

**Acknowledgments:** We acknowledge the National Council of Science and Technology of Mexico (CONACYT) for awarding a scholarship to the first author. We thank Hugo Zepeda for his valuable contribution and unstoppable support to validate and provide insight into the evidence needed for the present paper. Ana Perusquia took care of reviewing and improving the English version of the present manuscript.

**Conflicts of Interest:** The authors declare no conflict of interest.

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
