# Peer review of "Participatory Landscape Conservation: A Case Study of a Seasonally Dry Tropical Forest in Michoacan, Mexico"

_land, doi:10.3390/land12112016_

Round 1
Reviewer 1 Report
Comments and Suggestions for Authors
In this manuscript, “Participatory landscape conservation: a case study of a seasonally dry tropical forest in Michoacan, Mexico”, the authors apply an approach that involves local stakeholders in the decision of establishing a Biosphere Reserve. A comprehensive series of assemblies and workshops took place, gathering representatives from governments, academia, agrarian communities, etc. An area was chosen to establish the Zicuirán-Infiernillo Biosphere Reserve in 2005, with data collected in this year as the baseline. Data collected in 2021 was used for comparison purposes.
The manuscript provides a practical example of a participatory approach that is becoming increasingly relevant. It provides interesting details and conclusions that can be used to further encourage the use of this approach. However, there are several areas where the manuscript needs improvement:
⦁ Results.
Section 3.3 about the Biosphere reservemodel efficiency provides some interesting results regarding the established biosphere, and how the land use types have changed during the research years.
However, the article is missing some further insights about the reasons for these changes:
- Why did the cultural lands change of use? Who promoted this change? Was this change in land use supported in any way (financially or any other)? If so, who supported it? (there is a brief mention to this in Line 314, but more detail is needed with regards to which is the organization giving the incentives, and what kind of incentives were given to the communities; as well as what type of activities were being done by the communities to promote the recovery of the forest, and why they did decide to implement them).
- How did the land use change affect the stakeholders? Were the agrarian communities OK with the change? Did this change affect their economic income?
Based on Figure 3, even though the Recovery areas seem to outnumber the Disturbance areas (as supported by the data provided in Figure 2), it would still be relevant to explain why did the Disturbances occur within the Biosphere area. Where these Disturbances allowed or known by the Biosphere managers?
It would also be relevant to analyse the social component of the establishment of the Biosphere. It is well explained how the Biosphere was established in agreement with all the affected stakeholders, so it can be assumed that where the Biosphere was established, these stakeholders were OK with it. But what happened with the years?:
- Did the same stakeholders continue to be satisfied with the establishment of the Biosphere in 2021?
- Has the establishment of the Biosphere affected their lives in any way? If so, how?
- Has the Biosphere, and the process used to establish it, affected in any way, postively or negatively, the territory and the social disputes that were occurring in the area?
⦁ Discussion.
The results provided focus on an environmental aspect, the fact that the area of forest increased, vs. the land use area that decreased. This is a very relevant result. However, it would be relevant to provide further results, especially in the social area, as explained above in the 'results section'.
The participatory approach required a huge effort to be able to involve all the affected stakeholders. In order to justify this effort in future cases, further results and positive conclusions would be very helpful.
The discussion does make reference to other Biosphere Reserves that were established following a more top-bottom approach, which resulted in not-so-effective reserves. However, more details of this comparison is needed. How is the participatory approach to establish a Biosphere Reserve better than the traditional approach? Are there results available to support this conclusion and comparison? Are there any other details that could be provided in this regard?
The examples mentioned are interesting but more details are needed to make the case stronger about the potential and benefits of the participatory approach. For instance, in Lines 379-384 another Reserve is mentioned, where current disputes persist due to local actors not being on board with the original initiative. It would be relevant to include more information about this, and compare the results with the ZIBR reserve - have disputes occurred in the ZIBR reserve since its establishment? If so, what kind of disputes, and could these have been avoided by doing something differently in the participatory approach? If not, is this because of the comprehensiveness of the participatory approach, and the fact that a consensus was reached by all actors involved before establishing the ZIBR reserve?
Lines 399-404: as with the example above, more detail would be helpful to understand the example provided, and how the results of this research in Baja California supports the conclusions of the article. Two communities are mentioned in Baja California, but did both have the same results? It is an interesting example but not fully aligned with the purpose of the article. Was a participatory approach used in Baja California as a result of the communities being more in contact with travelers and tourist agencies? Why was the digital atlas created by engaging the community? Which organization promoted that engagement? Which have been the benefits for the community of this digital atlas? Is the 'resource' mentioned in Line 404 the digital atlas?
(Please, read the input in the section 'other issues' below for further detail about these examples)
⦁ Conclusions and next steps – what do the authors suggest doing with the conclusions of the article?
The article is lacking some recommendations or next steps to move forward. Given the positive results, and the increasing relevancy of participatory approaches, it would be necessary for the authors to include some or suggestions to move forward. How could these conclusions be used further? Are they planning on attending any events to present the results of the article? Or to meet with governments or NGOs representatives to share the results obtained?
Also, has any level of the goverment shown interest in adopting this participatory approach in other regions of the country?
Have the authors considered any next steps to promote this participatory approach? Are there any indications of the participatory approach being used in other regions or countries nearby?
⦁ References.
The participatory landscape approach used and explained in the article shares many elements with the Collective Impact Model.
The Collective Impact is a model developed by the Kennedy School at the Harvard University, which was published by the Stanford Social Innovation Review in 2011. It includes Five Conditions of Collective Success, very similar to those established in the article approach.
It is important for the authors to research this Collective Impact and analyze whether some synergies from both approaches could be joint together (to join forces in supporting the implementation of participatory approaches), and/or consider a mention to the Collective Impact in the references section.
⦁ Other issues
⦁ Line 81: what does 'Pas' refer to?; there is a typo: “…as a mean TO conserve....”.
⦁ Line 121: typo: "...it was referred TO as an outstanding..."
⦁ Line 135: "...three TYPES OF stakeholders---".
⦁ Line 139: "...on THE basis of...."
⦁ Line 258: ".... Numbers" Should be in singular.
⦁ Line 273: What does 'the other submissions' refer to? The establishment of the reserve in other areas. This is not clear, it would be useful to further explain the sentence.
⦁ Line 278: 'agreement instrument' seems to refer to 'means of agreement' in Appendix A1. Maintain the wording to avoid confusion.
⦁ Line 370: '...seems TO BE offering...'
⦁ Line 384: as explained above, the references to other examples are missing information when comparing with other reserves established without the participatory approach. The sentence 'relies upon a consensual governance regime' is not self-explanatory. Do the authors mean that the management of this reserve needs to be revised, by applying a participatory approach to reach a solution for a better management and persistence of this reserve? Are there any indication that a participatory approach will be implemented in this reserve? Are the authors actively doing anything in this regard, to support the improvement of this reserve?
⦁ Line 395: whoul would be trained? trained on what? how would that training help handle the disputes? which kind of disputes? How does this example relate to the article?
Comments on the Quality of English Language
In general, there are sentences that are a bit hard to follow, or cases where there is little detail provided, which makes it a bit complex to understand. It might be useful to provide a summary of the participatory approaches implemented, in a schematic way.
Some specific comments are provided in the section above.
Author Response
We are grateful for the comments and suggestions made by the reviewer. These help us to improve our present manuscript and most importantly to elaborate more on the relevance of visualizing our outcomes.
Specific details for all corrections are given in the attached file.

Reviewer 2 Report
Comments and Suggestions for Authors
Manuscript required minor revisions only.
Add no 1 for introduction
Add references- L-31, L-36,
In L-44 you have written like 'half of the all PA'. Is that across the world or any region? Please specify that.
L -81. Correctly spell out the acronym.
Use numerals for numbers above ten throughout the manuscript.
Please mention the X and Y axes in fig.2 and the yellow color for the bars representing tropical dry forest seem unattractive.
Try to avoid starting a sentence with the author's name, as this is often noted in the discussion part.
In reference, some of the references are given with pp to denote page number and some are not. Make them uniform.
Line- 124. It is better to use ‘-’ in between two years to represent a period. The authors have used both ‘-’ and ‘and’. Please remove ‘and’
Please follow a uniform way in writing the name of the state. Please stick on to ‘Michoacán’ or ‘Michoacan’
Line.238. Add a full stop after ‘et al’
Line. 242. Add full stop after ‘participate’
Make fig. 4 attractive and colourful.
L 331-334- Split the sentence.
Shift L.405 down to next page.
You could have added a conclusion part
Author Response
We acknowledge the comments and suggestions made by the reviewer. Specific details for all corrections are given in the attached file.

Reviewer 3 Report
Comments and Suggestions for Authors
The authors tackle a very relevant, useful and necessary area of inquiry. The experience presented acts as a demonstration project and the number of participants and participatory dynamics are impressive. The approach and results of the process are adequately designed, exposed and assessed. Maybe there is a lack of more specific results to enhance the quality and significance of the results, but the results integrated are enough.
In my view, to enhance the quality of the manuscript, here is a need of complementary information to fully comprehend the results of the study:
X In the first paragraph, the authors should give a synthesis of the causes that explain the failures to prevent environmental degradation in some protected areas.
Further on in the text, the authors state that there is a need to new approaches to better conserve the protected socio-ecological systems. Again, I think it would be insightful to make a synthesis of the main social conflicts affecting the functioning of these areas.
Finally, in the results and discussion there is a slight description of the situational context of social disputes in the study area, which is the core issue in my view. I acknowledge that the social disputes and the presence of violent groups make it difficult to describe in depth the conflicts. But I think that more details about that are needed, at least, to describe the nature of the conflicts existing in the analysed areas. This last description would be the conclusion derived from the previous argumentation around social issues in landscape conservation. It would be desirable to fully comprehend the context and the relevance of reaching an agreement between the social groups.
X I miss an analyse of the objectives, conservation approach and special conditions of a Biosphere Reserve, which is one of the conservation status analysed and established. Maybe when there is a reference to National Praks (line 74), a brief description of what a Biosphere Reserve is would be clarifying.
X The description of the study area lacks more details about the biophysical conditions or environmental values, the institutional management system and, importantly, about the social context. There are some ideas spread along the text that should be introduced in the Study area section.
The section Geopolitical context is very interesting, but I do not see a clear connection to the rest of the study. It should be synthesised and maybe integrated in the Study area section.
Finally, I would like to express my recognition to the participatory process conducted.
Author Response
We appreciate the comments and suggestions made by the reviewer. These help us to improve our present manuscript. Specific details for all corrections are given in the attached file.

Round 2
Reviewer 1 Report
Comments and Suggestions for Authors
Thanks to the authors for the thorough revision of the article. Some specific comments follow:
- Line 44: rather than 'overseen', the sentence refers to engagement of local stakeholders not being included, so overseen is not the right adjective. Please change for neglected; ignored or not included.
- Line 76: "...have addressed THIS PRINCIPLES, THE polycentric governance (....) and THE collective impact....".
- Line 77: " The FIRST ONE states...."
- Line 79: "The LATTER focusES on identifying....".
- Line 86: check the use of overseen is the intended one (see comment above).
- Line 87: does the reference to national park refers to 'strict' national parks, or to the areas within the national parks (Biosphere Reserves) referenced in Line 55 where local livelihoods are allowed?
- Line 135: "Climate varies DEPENDING ON THE ELEVATION or THE position...".
- Line 280: are the grey polygons protected areas that exist already? If so, please change from 'present' to 'current' or to 'already existing protected areas'. Same for the word 'present' in line 284.
- Line 359: it should say NEGOTIATION.
- Line 446: what Control does the sentence refer to? Control by the criminal organizations? Are these organizations against agricultural and exxtensive livestock and so they promote the maintenance of the native forests? Please clarify this.
Comments on the Quality of English Language
Please, see comments in section above.
Author Response
Please see the revised version. Please find the text highlighted in yellow as the suggestions from the reviewer. In red are the improvements of the English and editorial support.